# From Bench to Greenhouse: The Comparative Nano-Bio System Effects of Green-Synthesized TiO_2_-NPs and Plant-Growth-Promoting Microorganisms in *Capsicum annuum*

**DOI:** 10.3390/plants14233672

**Published:** 2025-12-02

**Authors:** Atiya Bhatti, Jorge L. Mejía-Méndez, Soheil S. Mamhoud, Araceli Sanchez-Martinez, Gildardo Sánchez-Ante, Jorge Manuel Silva-Jara, Eugenio Sánchez-Arreola, Luis Marcelo Lozano, Gonzalo Tortella, Edgar R. López-Mena, Diego E. Navarro-López

**Affiliations:** 1Tecnologico de Monterrey, Escuela de Ingeniería y Ciencias, Ave. General Ramon Corona 2514, Zapopan 45138, Jalisco, Mexico; a01783227@tec.mx (A.B.); gildardo.sanchez@tec.mx (G.S.-A.); marcelo.lozano@tec.mx (L.M.L.); 2Tecnologico de Monterrey, Escuela de Ingeniería y Ciencias, Epigmenio González 500, San Pablo 76130, Querétaro, Mexico; mejia.jorge@tec.mx; 3Department of Biology, The University of British Columbia, Okanagan Campus, 1177 Research Road, Kelowna, BC V1V 1V7, Canada; soheil.mamhoud@ubc.ca; 4Departamento de Ingeniería de Proyectos, CUCEI, Universidad de Guadalajara, Av. José Guadalupe Zuno # 48, Industrial los Belenes, Zapopan 45157, Jalisco, Mexico; araceli.sanchez46@academicos.udg.mx; 5Departamento de Farmacobiología, CUCEI, Universidad de Guadalajara, Blvd. Marcelino García Barragán 1421, Olímpica, Guadalajara 44430, Jalisco, Mexico; jorge.silva@academicos.udg.mx; 6Departamento de Ciencias Químico Biológicas, Universidad de las Américas Puebla, Santa Catarina Mártir s/n, San Andrés Cholula 72810, Puebla, Mexico; eugenio.sanchez@udlap.mx; 7Centro de Excelencia en Investigación Biotecnológica Aplicada al Medio Ambiente (CIBAMA-BIOREN), Facultad de Ingeniería y Ciencias, Universidad de La Frontera, Temuco 4811230, Chile; gonzalo.tortella@ufrontera.cl

**Keywords:** green synthesis, nanotechnology, precision agriculture, titanium dioxide nanoparticles, plant-growth-promoting microorganisms

## Abstract

In this study, titanium dioxide nanoparticles (TiO_2_-NPs) were produced via green routes using blueberry extracts obtained with isopropanol (I-TiO_2_-NPs) and methanol (M-TiO_2_ NPs). HPLC-DAD confirmed phenolic/flavonoid profiles in the extracts, and spectroscopy/microscopy established anatase, polyhedral, mesoporous TiO_2_-NPs with Eg ≈ 3.0 eV, hydrodynamic sizes ≈ 130–150 nm and negative ζ-potentials (−33 to −50 mV). The in vitro compatibility between TiO_2_-NPs and the plant-growth-promoting microorganisms (PGPMs) *Bacillus subtilis* (Bs), *Bacillus thuringiensis* (B), and *Trichoderma harzianum* (T) sustained increased growth up to 150 µg/mL without visible negative effects. In greenhouse experimentation of *Capsicum annuum* exposed to low-moderate TiO_2_-NPs, an increased leaf number and plant height were observed, while root length did not exceed the controls. I-TiO_2_ at moderate concentrations, particularly with a single PGPM (B or T), promoted fresh and dry biomass accumulation. Biochemically, peroxidase rose sharply for M-TiO_2_ at a low dose with consortium, whereas I-TiO_2_ elicited broader antioxidant responses; total protein increased at higher doses for both formulations, and total chlorophyll was highest with I-TiO_2_ (high dose with or without PGPMS). Collectively, the nano-bio system shows a formulation- and dose-dependent biphasic behavior: (I) I-TiO_2_ enhances biomass and photosynthetic pigments; (II) M-TiO_2_ favors strong POX induction under specific microorganism-dose combinations; and (III) single PGPM co-application with I-TiO_2_-NPs or M-TiO_2_ NPs outperforms consortia under our experimental conditions. Green synthesis thus provides surface functionalities that improve dispersion, microbial compatibility, and predictable physiological/biochemical outcomes for precision agriculture.

## 1. Introduction

Agriculture, the basis of food security and economic growth in many developing regions, is facing increasing pressure due to population growth, the decline in arable land, and adverse environmental conditions associated with climate change [1]. Nanotechnology is science, engineering, and technology conducted at the nanoscale, dealing with materials and structures at dimensions between approximately 1 and 100 nanometers. At this scale, materials can exhibit unique properties compared to their bulk counterparts, where their size, shape, or surface charge enable their interaction at a molecular or cellular level with components from various organisms [2]. The integration of nanotechnologically based approaches in agriculture occurs in precision agriculture, a distinguished innovative field focused on improving agricultural practices through optimized management, with the capacity to reduce waste, costs, and environmental impact, while enhancing data collection, timely interventions, crop yield, and sustainability [3].

The properties of nanomaterials are mainly conferred by the methods by which they are synthesized, ranging from an entirely chemical process, to partial biological intervention (green synthesis), to a completely biological process. Green synthesis refers to the environmentally friendly production of nanoparticles (NPs) using biological resources, such as plants and microorganisms or their derivatives. Unlike other synthesis routes, green synthesis focuses on the use of non-toxic, renewable materials and mild reaction conditions to yield biocompatible, stable, and highly efficient nanostructures [4]. Like other sources, plant extracts are widely used in green synthesis due to their abundance, cost-effectiveness, and eco-friendly nature. However, they are preferred since they can contain complex mixtures of proteins, enzymes, polysaccharides, amino acids, phytochemicals, and vitamins that act as reducing and stabilizing agents during the synthesis of nanoscale materials [5]. *Vaccinium corymbosum*, commonly known as highbush blueberry, belongs to the family Ericaceae and the genus *Vaccinium*. The phytochemistry of *V. corymbosum* is notable for its high content of anthocyanins, flavonoids, and phenolic acids, which contribute to its potential health benefits and potential applications in the nanotechnological research field.

Among metal oxide-based nanomaterials, titanium dioxide nanoparticles (TiO_2_-NPs) stand out for their photocatalytic, UV-blocking, and redox-active properties, which enhance light absorption, stimulate photosynthesis, and regulate enzymatic activities essential for plant metabolism [6,7]. Thanks to these characteristics, TiO_2_-NPs have been associated with higher germination rates, increased biomass, and greater tolerance to abiotic stresses such as salinity, drought, and heavy metal toxicity [8,9,10]. Furthermore, TiO_2_ promotes nutrient uptake by enhancing the absorption of essential elements such as Fe, K, Mg, and Zn, thereby contributing to the development of healthier, more productive plants [11]. Beyond the agricultural sector, TiO_2_ is a material of high industrial and economic relevance, widely used as a pigment, in coatings, photocatalytic purification systems, and self-cleaning surfaces [12]. However, the conventional chemical synthesis of TiO_2_ involves the use of aggressive reagents and results in significant environmental impacts.

In response, green synthesis or synthesis mediated by plant extracts has emerged as an environmentally friendly and sustainable alternative, based on the use of phytochemical-rich plant extracts that act as reducing and stabilizing agents. This approach reduces toxicity and energy consumption and improves the biocompatibility, stability, and dispersibility of NPs thanks to the formation of natural organic coatings [13,14]. From a microbiological perspective, metal oxide NPs, including TiO_2_, have demonstrated antimicrobial activity against Gram-positive bacteria such as *Bacillus subtilis* and *Bacillus thuringiensis*, attributed to the generation of reactive oxygen species (ROS) under light exposure [15]. However, when TiO_2_ is obtained through green methods, this activity can be moderated, favoring mutualistic interactions with plant-growth-promoting microorganisms (PGPMs) that promote plant health and improved physiological and biochemical responses. Among them, *Trichoderma harzianum* is recognized as a biofertilizer and biostimulant that improves root development, nutrient uptake, and plant immune response [16,17].

Therefore, the green synthesis route was followed in this study for obtaining two formulations of TiO_2_-NPs based on isopropanol and methanol extracts from blueberries and PGPMs, as a novel alternative for improving the physiological and biochemical features of *C. annuum*. Given the use of solvents, samples were designated as isopropanol (I-TiO_2_) and methanol (M-TiO_2_), which were further utilized to evaluate their effect on the physiology and biochemistry of *C. anuumm* plants. The in vitro compatibility of I-TiO_2_ and M-TiO_2_ with PGPMs (*B. subtilis*, *B. thuringiensis*, and *T. harzianum*) was explored, together with their performance as a nano-bio system in antioxidant enzyme activity, accumulation of phenolic compounds, and total protein content in plants. This comprehensive approach seeks to elucidate the dual role of TiO_2_-NPs as a biocompatible nanomaterial and nanobiostimulant, contributing to the transition toward a more sustainable agriculture that combines productivity, resilience, and environmental responsibility.

## 2. Results

### 2.1. Synthesis and Characterization of Green-Synthesized TiO_2_-NPs

To investigate the crystalline structure and phase identification of green-synthesized I-TiO_2_ and M-TiO_2_ NPs, the XRD characterization technique was followed. The XRD patterns of I-TiO_2_ and M-TiO_2_ (Figure 1a) showed peaks in the (101), (004), (200), (105), (211), (204), and (110) planes, corresponding to the tetragonal anatase phase of TiO_2_ (JCPDS #21-1272), with no detectable secondary phases or impurities. Using the Scherrer equation, the average crystallite size was 12.91 ± 5% nm for I-TiO_2_ and 12.42 ± 5% nm for M-TiO_2_. The optical properties (Figure 1b) showed similar UV–Vis spectra for both materials, with a defined absorption edge around ≈350 nm. Using the Kubelka–Munk and Tauc equations, the optical bandgap (E_9_) was 2.98 eV for I-TiO_2_ and 3.01 eV for M-TiO_2_. The FT-IR spectra (Figure 1c,d) included, for the blueberry extracts (4000–400 cm^−1^), characteristic bands 3200–3600 cm^−1^ (ν-OH), ~2120 cm^−1^ (ν C≡C), and 1270 cm^−1^ (ν-CO) and signals at 660 and 870 cm^−1^ (δ C-H). In the extract + TTIP systems, bands were observed at 1643 cm^−1^ (ν C=O) and around 1030–1129 cm^−1^ (ν C-O); for the isopropanol system, isopropyl group bands were also detected at 1390–1365 cm^−1^. After calcination, the M-TiO_2_ and I-TiO_2_ solids showed bands at ~3291 cm^−1^ (adsorbed water) and ~1670 cm^−1^ (surface -OH), as well as signals at 800–400 cm^−1^, attributable to Ti–O/Ti–O–Ti bonds; in M-TiO_2_, peaks were identified at 545 and 745 cm^−1^, associated with ν(Ti–O). The spectrum of calcined I-TiO_2_ was similar to M-TiO_2_, with a lower moisture signal.

The morphological characterization of the M-TiO_2_ and I-TiO_2_-NPs was obtained as shown in Figure 2. High-magnification SEM micrographs (Figure 2a,b) show that both materials exhibit an interconnected polyhedral morphology. Statistical grain size analysis (Figure 2c) determined the average grain size to be 55.49 nm for M-TiO_2_ and 49.57 nm ± 5% for I-TiO_2_, with no significant differences between the two. Pore size and specific surface area were evaluated using the Brunauer–Emmett–Teller (BET) method, while pore distribution was determined using the Barrett–Joyner–Halenda (BJH) method (Figure 2d). The average pore diameters were 24.3 nm for M-TiO_2_ and 17.5 nm ± 5% for I-TiO_2_. The N_2_ adsorption/desorption isotherms (inset in Figure 2d) showed a type V curve according to the IUPAC classification, with an H2 hysteresis loop, associated with “ink-bottle” pores, with low connectivity and an irregular structure [18]. The calculated surface area was 46.51 m^2^ g^−1^ for M-TiO_2_ and 41.65 m^2^ g^−1^ for I-TiO_2_. The hydrodynamic size was 146.1 and 133.2 nm ± 5%, respectively, and the ζ potential was –50.6 mV for M-TiO_2_ and –33.37 mV for I-TiO_2_, indicating good colloidal stability and dispersion in the aqueous medium. Particle size dispersion (DLS) analysis confirmed homogeneous dispersion and good stability as suspended in the aqueous media for both samples, with average hydrodynamic sizes of 146.1 nm for M-TiO_2_ and 133.2 nm for I-TiO_2_, with no significant differences between them (Figure 2e). A small secondary fraction was also observed in M-TiO_2_ around 5468 nm, attributable to slight particle aggregation or the formation of transient agglomerates in suspension. Zeta potential values were obtained: –50.06 mV for M-TiO_2_ and –33.37 mV for I-TiO_2_. Both indicate a negative surface charge and high colloidal stability, since magnitudes greater than |30| mV reflect strong electrostatic repulsion between particles. However, the more negative value observed in M-TiO_2_ suggests a greater electrostatic stabilization, probably associated with a higher density of hydroxyl groups (–OH) or functional residues of the methanolic solvent on the surface (Figure 2f).

### 2.2. HPLC-DAD Analysis of Extracts from Blueberry

According to the implemented method, the compounds identified in the isopropanol extract from blueberry by HPLC-DAD are listed in Table 1. The chromatograms are illustrated in Appendix A. Considering the available literature, the phytochemical composition of the methanol extract is presented in Appendix A.

According to the data presented in Table 1, the synthesis of TiO_2_ in the presence of organic compounds proceeds in two main stages. First, titanium(IV) isopropoxide is initially hydrolyzed, with the isopropoxide groups gradually replaced by OH groups, releasing isopropanol as a byproduct. These hydroxylated species undergo oxo- and alkoxy-condensation reactions, forming Ti-O-Ti bonds and generating an inorganic network that serves as a precursor to the metal oxide [19]. Notably, the extract utilized in the synthesis process comprises a complex matrix of polar-based compounds that act synergistically during the reduction and stabilization of the titanium precursor into its nanoparticulate form [20]. In this regard, the process is significantly modified because the phenolic compounds contained in this matrix, such as flavonoids, can coordinate with Ti(IV), stabilizing intermediates and facilitating the hydrolysis of the precursors through proton donation and the activation of Ti-OR bonds [21]. In turn, for flavonoids, it can be hypothesized that they are progressively oxidized until they decompose completely into CO_2_ and H_2_O, while the titanium hydroxide network is finally transformed into crystalline TiO_2_ [21].

### 2.3. Analysis of the Compatibility Between Green-Synthesized TiO_2_-NPs and PGPMs

The biocompatibility of M- and I-TiO_2_-NPs with PGPMs was analyzed; Figure 3 shows the compatibility and growth response of PGPMs against I- and M-TiO_2_-NPs. First, it was observed that in solid medium, that is, on the plates, *B. subtilis* and *B. thuringensis* microbial growth gradually increased with concentration, reaching the highest values between 100 and 150 µg/mL, without showing visible inhibitory effects. This confirms that the NPs do not generate toxicity; on the contrary, they maintain or stimulate bacterial growth, especially with M-TiO_2_ (Figure 3a,b). Regarding beneficial fungi, in the case of *T. harzianum*, it is observed that mycelial growth was greater with I-TiO_2_, as was the number of spore-forming units, reaching a positive effect at intermediate concentrations, with no evidence of inhibition or morphological alterations (Figure 3c).

### 2.4. Plant Physiological Response of TiO_2_-NPs and PGPMs

As illustrated in Figure 4, the M- and I-TiO_2_-NPs interaction showed a significant effect (*p* < 0.05) across various treatments (an illustrative process of the Nano-bio system is shown in Figure 4a). The NPs induce a remarkable effect on plant height. The results show that the most effective treatments correspond again to what was observed in the leaf; TiO_2_ nanoparticles demonstrated the best results at 150 μg/mL (C2), both for M-TiO_2_ (22.7 ± 5.9 cm) and I-TiO_2_ (22.5 ± 3.8 cm), compared with the control (20.06 ± 5.8 cm), especially when applied alone or in combination with the single microorganism *B. subtilis* or *T. harzianum* (Figure 4b). In contrast, low doses (C1) and treatments with the microbial consortium (Mcg) showed the weakest heights (letters c–e), indicating a possible inhibition associated with excess NPs or complex microbial interactions.

Regarding the increase in the number of leaves (Figure 4c), the best treatments were those corresponding to TiO_2_ at 50 μg mL^−1^ (C1), both for M-TiO_2_ and I-TiO_2_, in their performance when Np acted alone or in combination with one of the microorganisms (*B. subtilis* or *T. harzianum*), ranging from 13.5–16.5 leaves in contrast with the control (7.7 ± 1.28 leaves). These treatments are located in the upper group (letter “a”/”ab”) and significantly outperform the control, showing the highest average number of leaves, ranging from 14 to 16 ± 2.3 leaves. Unlike the with pure NPs of the same dose (C1), adding *B. subtilis* or *T. harzianum* did not provide a statistically significant improvement. The microbial consortium is not among the best results and does not outperform pure NPs.

Regarding the function of root growth promoters, no increase in root length was observed compared to the control (Figure 4d). TiO_2_-NPs at 50 μg/mL (C1), especially when combined with a single microorganism (*B. subtilis* or *T. harzianum*), maintained a similar behavior to the control. Unlike what was observed in the number of leaves, pure NPs did not outperform the control, and in some cases root length even tended to decrease slightly at high concentrations (C2) or with the microbial consortium (Mcg), which appear in groups c–e. Overall, the application of TiO_2_ at low doses, alone or with a single microorganism, maintained a response comparable to the control, while multiple combinations or high concentrations reduced root length.

For the fresh weight variable, the results show distinct behavior between the two types of NPs. In the case of M-TiO_2_, no significant differences were observed between treatments; all groups, including pure NPs and combinations with microorganisms, were statistically within the same range as the control (2.17 ± 0.49). In contrast, more pronounced differences were observed for I-TiO_2_: treatments C2 (5.4 ± 0.41), C2-T (5.17 ± 0.53), and C2-B (5.23 ± 0.53) ranked among the highest (letters a–ab), significantly exceeding the control, which had the lowest average (letter e). This indicates that, in this case, TiO_2_ synthesized with isopropanol at low and medium concentrations—especially when combined with individual microorganisms—promoted the accumulation of fresh biomass. Conversely, the consortia and higher concentrations showed more variable responses or no clear increase (Figure 4e). To validate biomass or efficient water absorption capacity, dry weight was quantified. In this case, the M-TiO_2_ treatments did not show significant differences between themselves or with respect to the control (1.21 ± 025 g); all were located within the same statistical group (letter “a”), indicating a uniform response without significant alterations in dry biomass accumulation. This coincides with the behavior observed in fresh weight, where no marked variations were present either. In contrast, the I-TiO_2_ treatments exhibited greater variability and a more sensitive response to the combination with microorganisms. Treatments C (3.3 ± 0.46 g), C2-T (2.56 ± 0.37), and C2-B (3.07 ± 0.39) showed the highest dry weight values (letters a–ab), surpassing the control (letter c), and replicating the increasing trend observed in fresh weight. This shows that TiO_2_ synthesized with isopropanol promoted both total biomass and the preservation of dry mass, especially in simple combinations with individual microorganisms (Figure 4f). Both parameters confirm an increase in structural biomass.

According to the plant phenotype observations illustrated in Figure 5, a visual comparison of plants exposed to the different treatments clearly supports the quantitative results described above. The interaction between TiO_2_-NPs and beneficial microorganisms generated a dose- and formulation-type-dependent response in plant growth and morphology. The most vigorous plants, with greater leaf area, stem elongation, and good turgor, corresponded to treatments with low concentrations (50 μg/mL) of M-TiO_2_ and I-TiO_2_, applied either alone or in combination with a single microorganism (*B. subtilis* or *T. harzianum*). These plants displayed denser foliage and more developed root systems compared to the control. Notably, the I-TiO_2_ formulations promoted a more uniform and robust phenotype, particularly in treatments C1 and C2 combined with individual microorganisms, which coincides with the observed increase in fresh and dry biomass. Meanwhile, the M-TiO_2_ treatments maintained stable growth, comparable to the control, suggesting that methanolic synthesis did not generate stress in the plants, although it produced more conservative physiological responses.

### 2.5. Biochemical Response of TiO_2_-NPs and PGPMs

The metabolic response regarding antioxidant markers, chlorophyll, and total protein to NP exposure was observed (Figure 6). Peroxidase activity (AU/g FW) showed clear differences between treatments, mainly associated with the type of nanoparticle and its combination with microorganisms (Figure 6a). In the case of M-TiO_2_, the highest activity was recorded in treatment C1-M (12.7 ± 4.3), corresponding to a low dose (50 μg/mL) combined with microorganisms, which reached the maximum value and was located in group “a,” significantly higher than the rest. The remaining treatments, including the control and pure NPs, were concentrated in group “b,” with lower and similar levels of peroxidase activity. This suggests that, in the methanolic formulation, enzyme induction was specific and dependent on the interaction between NPs and microorganisms rather than on the direct effect of TiO_2_. For I-TiO_2_, the trend was more widespread: treatments C1-M (10.009 ± 1.87 UA/FW), C2-B (7.3 ± 1.39 UA/FW), and C2-M presented significantly higher values (groups “a” or “ab”) compared to the control. This pattern indicates that I-TiO_2_ promoted a broader antioxidant response, with moderate but consistent peroxidase activation across multiple treatments, particularly when combined with individual microorganisms. Overall, the results show that peroxidase activity is enhanced in the presence of NPs and microorganisms, but the most pronounced effect occurs in the consortium combination (C1-M), especially with M-TiO_2_, suggesting a potential synergy between the nanoparticle and the activity of the microorganisms, which stimulates oxidative defense enzymatic pathways.

Regarding the total phenolic compound content (nmol GA/g FW), notable differences were observed between treatments and nanoparticle types (Figure 6b). In the case of M-TiO_2_, the control (C) presented one of the highest values, while most of the nanoparticle treatments—including C1, C1-T, C1-B, and C2-T—showed intermediate levels (b–c), without surpassing the control. Only treatment C2-B statistically approached the highest level (a–b), suggesting a limited response in the induction of phenolic compounds under the methanolic formulation. In contrast, the I-TiO_2_ treatments exhibited a more defined increase pattern, with C2-B (44.65 ± 0.071 nmol GA/g FW) standing out, reaching the highest value (letter “a”), significantly higher than the control (35.47 ± 2.61 nmol GA/g FW) and the rest. Treatments C2-B and C1-M also showed relevant increases (b–c), while treatments with the microbial consortium (Mcg) and individual microorganisms without NPs (B and T) presented the lowest values (d–f).

The effect of NPs also generated a significant increase in total protein content (Figure 6c). Marked differences were observed between treatments and formulations. In the case of M-TiO_2_, treatments C2, C2-T, C2-B, and C2-M reached the highest values (from 117–218 µg/g FW), significantly exceeding the control and low doses. This suggests that high concentrations of NPs, especially when combined with microorganisms, stimulate protein synthesis or accumulation, potentially associated with the activation of secondary and defense metabolism. Low-concentration treatments (C1, C1-T, C1-B, C1-M) maintained intermediate levels, while the control and microorganisms without NPs recorded the lowest values. A similar pattern was observed with I-TiO_2_, where the highest concentrations (C2-T, C2-B, C2-M) also produced the highest levels of total protein (group “a”), outperforming the control and treatments with NPs alone or with individual microorganisms at low doses.

For the total chlorophyll variable, clear differences were observed between treatments and nanoparticle types (Figure 6d). For M-TiO_2_, values were moderate and consistent: treatments with NPs, both alone and combined with microorganisms, were grouped in a–c, while the control showed the lowest levels (d–f). This indicates that methanolic synthesis of TiO_2_ caused a gentle and stable increase in chlorophyll content, without significant spikes. In contrast, I-TiO_2_ showed a much stronger response, especially in treatment C2, which reached the highest value (1012.46 ± 70.9), followed by C2-B and C2-M (783.48 ± 87.43, and 879.35 ± 46.13). These findings suggest that the isopropanolic formulation (I-TiO_2_ NPs), particularly when combined with microorganisms, enhances the biosynthesis of photosynthetic pigments and may improve the efficiency of the photosynthetic apparatus.

The mechanisms by which plant development is improved involve physiological and biochemical modifications; I-TiO_2_ and M-TiO_2_-NPs promoted a biphasic effect dependent on the concentration and the type of associated microorganism (Figure 7). At moderate concentrations (50–150 μg/mL), the NPs favored mycelial growth and sporulation of *T. harzianum*, while no inhibitory effects were observed in *B. thuringiensis*, confirming the compatibility of the nano-bio system. These results suggest that TiO_2_-NPs enhance plant growth, potentially by redirecting hormonal flow, improving metabolic efficiency, and maintaining cellular integrity through enhanced oxygen availability, redox modulation, and the activation of antioxidant enzymes.

### 2.6. PLS Analysis

Interestingly, PLS analysis identified the nanoparticles’ physical bioavailability as the primary determinant of the biological effect observed in plants. Among the properties evaluated, hydrodynamic size (DLS) and primary crystal size (PS) showed the highest importance values (VIP > 0.75), indicating that they are the physicochemical variables with the greatest contribution to modulating physiological and biochemical responses (see Appendix A).

## 3. Discussion

Green synthesis represents a safe and environmentally friendly approach to the production of nanomaterials for precision agriculture. In precision agriculture, TiO_2_-NPs play a significant role by enhancing crop productivity and sustainability by enhancing seed germination, plant growth, crop yield, and physiological and biochemical parameters. Contrary to top-down-based NMs, green-synthesized TiO_2_-NPs can exert multiple biological effects among crops due to the presence of coating agents, uniform distribution, controlled particle size, and compatibility with PGPMs. Still, the biological performance of TiO_2_-NPs can also vary depending on the utilized reducing agent and its polarity.

HPLC is an analytical technique used to separate, identify, and quantify components in a mixture. For green synthesis approaches, HPLC is required for identifying phenolic acids or flavonoids among polar extracts. Here, HPLC analysis was utilized to identify potential reducing agents in the blueberry extracts. The results demonstrated the presence of flavonoids and phenolic acids in the isopropanol extract. These compounds consisted of rutin, quercetin, hesperidin, and 3,5-dihydroxybenzoic acid. Regarding the available literature, it was noted that the methanol extract is conformed by catechin, chlorogenic acid, and the glycosidic derivatives of petunidin and malvidin (see Appendix A). The determined compounds are consistent with other studies that have been implemented for evaluating the phytochemistry of blueberries. For instance, a recent study unveiled by HPLC/MS showed that blueberry extracts obtained via ultrasound-assisted extraction contain acetylated glucose moieties as well as quercetin, myricetin, kaempferol, and syringetin [22]. The same work mentioned the presence of sugars such as rhamnoside, pentoside, and rutinoside. Similarly, other studies that have included HPLC-DAD-ESI-MS^n^ evaluation of blueberry varieties (e.g., bluegem, climax, and powderblue) have demonstrated that they are abundant sources of chlorogenic acid, laricitrin, and flavonol glycosides [22]. The content of phenolic acids has also been unveiled in blueberry extracts prepared with methanol, where recent studies have reported that they can contain gallic acid, catechin, caffeic acid, vanillic acid, and protocatechuic acid [23]. The variabilities between the compounds identified in this study with other reports can be associated with the column type, length, and diameter, together with the mobile phase composition, sample preparation, detection method, and standardization of the utilized instrument.

The characterization of green-synthesized NMs involves the comprehensive analysis of their physical, chemical, and biological properties through spectroscopy and microscopy approaches. For precision agriculture applications, the characterization of NMs is required for ensuring their safety, phytopathogen management, optimal performance in agricultural systems, and possible interaction with plants, soil, and the environment [24]. Here, it was noted I-TiO_2_ and M-TiO_2_ NPs exhibited characteristic bands at 332 and 334 nm related to the confirmation of the reduction of metallic ions into their nanoparticulated form. Together with this, FT-IR analysis demonstrated a series of bands that can be associated with the presence of the blueberry extract as a coating agent. The former event is of great importance, since it is associated with the surface plasmon resonance effect of the formed I-TiO_2_ and M-TiO_2_ NPs, which serves as indicator of stability, biocompatibility, and target capacity against potential phytopathogens. The latter phenomenon is significant, since it demonstrated that the presence of blueberry extract can aid in the prevention of agglomeration and enhanced uniform dispersion. The data retrieved from the UV–Vis and FT-IR spectroscopy analyses is challenging to compare since current studies have focused on utilizing blueberry extracts for obtaining TiO_2_-NPs with photocatalyzing applications [25], where it has been noted that TiO_2_-NPs can display an *Eg* of 3.3–3.4 eV. The *Eg* of NMs is a relevant feature to be evaluated, since it reflects their capability to release nutrients, generate ROS, and improve growth rate and crop yields. Despite this, it should be noted that the variabilities between the calculated *Eg* reported in other studies with the ones reported in this work (*Eg* 2.98–3.05 eV) can be related to the nature of the utilized precursors and the polarity of the obtained blueberry extract. The size, shape, and surface charge are other features that are challenging to compare with the existing literature regarding the green synthesis of TiO_2_-NPs with blueberry extracts. However, in studies involving the development of TiO_2_-NPs with extracts from *Aloe vera* [26] and *Calatropis procera* [27], it has been observed that they are significant features influencing the attenuation of salt stress, the incidence of dusky cotton bugs, and photosynthetic capacity.

PGPMs constitute a wide category of microorganisms that promote plant growth and health through various mechanisms. Among PGPMs, *B. thuringiensis* and *B. subtilis* are associated with the genus *Bacillus* within the phylum Firmicutes, while *T. harzianum* is a fungus classified under the genus *Trichoderma* in the phylum Ascomycota. Comparably to other microorganisms considered in precision agriculture approaches, *B. thuringiensis*, *B. subtilis*, and *T. harzianum* can improve plant growth and health through various mechanisms. For instance, *B. subtilis* has been documented for its capability to promote plant growth by producing phytohormones, solubilizing nutrients, and inducing systemic resistance against pathogens [28]. Similarly, *B. thuringiensis* has been noted for its beneficial features in contributing to soil microbial diversity and activity, root development, plant vigor, and stress resistance [18]. In the same context, *T. harzianum* has been reported for its capacity to induce systemic resistance, mycoparasitism, and symbiotic relationships that enable nutrient uptake and nutrient solubilization. Regarding their compatibility, it was observed that treatment with I-TiO_2_ and M-TiO_2_ NPs did not compromise the CFU nor SFU capacity of Bt, Bs, and Th at 50, 100, and 150 mg/mL. The compatible effect of I-TiO_2_ and M-TiO_2_ NPs with the PGPMs cultured in this study can be related to the capability of TiO_2_-NPs to influence the pH of the growth medium, which can enhance the metabolic activity and favorable growth of Bt, Bs, and Th. The compatible effect can also be associated with the presence of the blueberry extract onto the surface of I-TiO_2_ and M-TiO_2_ NPs, where its content of flavonoids and phenolics acids can serve as a nutrient source for Bt, Bs, and Th, enhancing their growth rate, colony formation, and spore production [29]. The presence of the mentioned organic layers on the surface of I-TiO_2_ and M-TiO_2_ NPs can be associated with the results retrieved from FT-IR analysis, where M-TiO_2_ NPs exhibited characteristic bands from the utilized berry extract. In the case of I-TiO_2_ NPs, the existence of an organic layer on their surface can be related to the variabilities between the ζ-potential determined in this work (−33.37 mV); previous studies from our research group demonstrated that TiO_2_-NPs synthesized by a chemical method can exhibit larger negative surface. From an experimental setting perspective, the calculated ζ-potential can also reflect the presence of organic layers on I-TiO_2_, since they might be modifying the refractive index of the synthesized NPs in the medium and hence yield different ζ-potential values. Additionally, we envisioned the presence of the mentioned organic layers during the reduction process rather than on the final product.

The ζ-potential of I-TiO_2_ (−33.37 mV) and M-TiO_2_ NPs (−50.06 mV) can be also related to the upregulated capacity of Bt, Bs, and Th in enhancing CFU and SFU, since it can be hypothesized that their negative surface charge can enable superior microbial attachment due to their interaction with positive charged molecular components. The result of this process can occur in enhanced biofilm formation, microbial stability, and survival upon exposure to treatment. The influence of the ζ-potential of NMs has been mainly documented in the biomedical research field, where it has been considered that NMs with ζ-potential ranging from −30 to +30 mV tend to exhibit high stability and ideal biological performance [30]. Still, in the precision agricultural pipeline, this evidence is scarce in current studies, as based on the literature review. Thus, further studies are required to validate this.

The physiological parameters of a plant refer to the measurable indicators of its internal biological processes and functions. Representative physiological parameters encompass leaf number, root length, plant height, and fresh and dry weight. When nanotechnology is integrated into precision agriculture, it is necessary to evaluate the leaf number since it is associated with increased light interception, photosynthetic capacity, biomass production, and plant vigor. In the same regard, root length must be determined, since it is associated with the ability of crops in accessing water and nutrients, where longer root systems can enhance the resilience of plants to abiotic stresses. The plant height is another feature frequently evaluated when NMs are considered, since it is an effective indicator of growth performance and vigor, which reflects the capability of plants to compete for light, signal stress, and nutrient deficiencies.

The combined effect of TiO_2_-NPs and plant growth-promoting microorganisms (PGPMs) showed a clear dependence on the type of synthesis, surface structure, and applied dose, reflecting a complex interaction between the components of the nano-bio system. According to the characterizations of the M-TiO_2_ and I-TiO_2_-NPs, they exhibited a pure anatase crystalline structure, with nanometric crystallite sizes (≈12–13 nm), an optical bandgap of 2.98–3.01 eV, and a mesoporous polyhedral morphology. These characteristics, reported to be determinants of the photocatalytic activity and biocompatibility of TiO_2_, explain the differences in the physiological response observed in plants [14]. The nanometric size and high specific surface area (≈42–46 m^2^ g^−1^) increase surface reactivity, favoring interaction with biomolecules and rhizospheric microorganisms, as well as the adsorption of water and essential nutrients [31]. In this context, the nanoparticles promoted significant increases in leaf height and number, especially at low doses (50–150 μg/mL), applied alone or in combination with *B. subtilis* or *T. harzianum*. These results are consistent with previous reports in which moderate doses of TiO_2_-NPs stimulate cell division, stem elongation, and leaf expansion through increased photosynthetic efficiency and carbon assimilation, attributed to TiO_2_’s ability to optimize light harvesting and electron transport in photosystems I and II [32].

The increase in fresh and dry biomass, more evident in the I-TiO_2_ formulations at 150 μg/mL, may be related to the lower surface moisture content and greater photocatalytic capacity of this formulation. The controlled generation of reactive oxygen species (ROS) by anatase TiO_2_ acts as a moderate redox signal that stimulates metabolic pathways associated with the biosynthesis of lignin, structural proteins, and secondary metabolites, contributing to greater plant growth and vigor [33]. In contrast, M-TiO_2_, with a higher negative charge (ζ = –50 mV) and a high density of hydroxyl groups (-OH), displayed more stable and biocompatible behavior, suggesting a more passive interaction with plant tissue and better compatibility with microorganisms without generating excessive oxidative stress that leads to uncontrolled programmed cell death [34]. Interaction with individual PGPMs enhanced the positive effects of the NPs. *B. subtilis* and *T. harzianum* are recognized for their ability to release phytohormones, siderophores, and hydrolytic enzymes, which increase nutrient absorption and root growth [35]. The mesoporous surface of the NPs, with pore diameters between 17 and 24 nm, likely facilitated the adsorption and progressive release of these bioactive compounds, generating a more efficient rhizospheric microenvironment. However, the use of microbial consortia (Mcg) showed less consistent results, potentially due to metabolic competition between strains or interference in colonization/interaction, an effect that could be amplified by the observed pore heterogeneity.

From a biochemical perspective, TiO_2_ acted as a modulator of antioxidant metabolism. POX activity was significantly increased in the presence of TiO_2_-NPs, especially in the methanolic formulation combined with microorganisms, demonstrating the activation of enzymatic defense mechanisms against ROS [36]. In plants treated with I-TiO_2_, the effect was more widespread, with sustained activation of POD and a parallel increase in total phenolic compounds, especially with *B. subtilis*, confirming that ROS photoinduced by TiO_2_ anatase act as physiological signals to activate redox and phenolic pathways [37]. Phenolic compounds play key roles in neutralizing ROS and regulating redox balance, reinforcing cellular integrity and the response to environmental stress. The increase in total protein content in treatments with high concentrations (C2) is associated with increased synthesis of structural and enzymatic proteins, linked to the activation of photosynthetic and defense metabolic pathways [38]. It has been described that nanoparticle–microorganism interaction can regulate the gene expression of nutrient transporters and antioxidant proteins, improving the metabolic efficiency of the system.

Biochemical parameters of a plant encompass a range of molecular and chemical characteristics that reflect its physiological state and metabolic processes. For precision agriculture purposes, frequently analyzed biochemical parameters include enzyme activity, phytochemical content, chlorophyll content, and protein content. Compared to other features, the analysis of the activity of enzymes such as POX is required since it is correlated with the capacity of crops to exert defense responses against phytopathogens, promote stress response, modify cell wall modifications to provide structural support, and maintain photosynthesis and energy metabolism. Similarly, the evaluation of the phytochemical content of crops upon exposure to NMs is necessary since they can serve as an indicator of the capacity of plants for displaying growth regulation, abiotic stress tolerance, lignification, and defense mechanism phenomena. For human health considerations, the phytochemical content of crops must be considered to assess their value as a source of anti-inflammatory, anticancer, or antidiabetic agents. Regarding protein content, it is a significant parameter to be recorded since it reflects the capability of the plant to orchestrate biochemical reactions as well as the integrity and development of cellular components.

Likewise, total chlorophyll accumulation was more pronounced in the I-TiO_2_ treatments, particularly in C2, C2-B, and C2-M, reflecting greater efficiency in light harvesting and energy conversion. This behavior is related to the high crystalline purity and narrow bandgap (~3.0 eV) of the NPs, characteristics that favor electronic excitation in the UV–visible range and stimulate Rubisco and ATPase activity, essential for carbon fixation [8]. Furthermore, the presence of PGPMs could have intensified the biosynthesis of photosynthetic pigments through the production of indoleacetic acid (IAA) and gibberellic acid (GA_3_), promoting greater leaf area and photosynthetic efficiency. Overall, the correlation between the physicochemical properties of green-synthesized TiO_2_ and the physiological and biochemical responses of plants demonstrates that mesoporous morphology, colloidal stability, and surface composition are determining factors in its biological performance.

Formulations with greater dispersion and smaller effective size (I-TiO_2_) were associated with more intense activation of redox signaling pathways, reflected in increased peroxidase activity and phenol content. Conversely, particles with a higher degree of agglomeration (M-TiO_2_) showed better performance in structural and photosynthetic parameters, thereby enhancing vegetative growth and biomass accumulation. Although PLS suggested that agglomerated materials (M-TiO_2_) associate with structural traits, the experimental results showed that highly dispersed nanoparticles (I-TiO_2_) delivered the most substantial improvements in plant growth and biomass. This suggests that the increased bioavailability and nano–bio interface efficiency of I-TiO_2_ enable a dual mechanism: mild ROS-driven signaling that primes metabolism, along with enhanced photosynthetic performance when combined with PGPMs. Therefore, I-TiO_2_ demonstrates superior functional performance as a nano-bio stimulant. From a design perspective, this shows that controlling the colloidal state and suspension stability are critical factors for directing the functionality of nano-bio systems.

Despite the provided evidence in this work, it is noteworthy to mention that the large-scale application of bio-based nanoparticle systems faces several challenges, including the complexity of synthesis methods, variability in nanoparticle characteristics, and potential toxicity concerns. Variations in the production process can lead to inconsistencies in size, shape, and surface properties, which are critical for their effectiveness in enhancing plant growth and productivity. Additionally, the scaling up of synthesis techniques may introduce economic limitations and regulatory hurdles, as well as the need for thorough safety assessments to address environmental and health impacts. To overcome these challenges, researchers should focus on standardizing synthesis protocols and employing quality control measures to ensure uniformity. Furthermore, interdisciplinary collaboration can facilitate a better understanding of nanoparticle interactions with biological systems, leading to the development of tailored solutions that are both effective and sustainable for agricultural applications.

## 4. Materials and Methods

### 4.1. Extract Obtention and Green Synthesis of I-TiO_2_ and M-TiO_2_

Preparation of plant extract: 50 g of Highbush Blueberry (*V. corymbosum*) was ground in an agate mortar. The pulp was then mixed with 50 mL of deionized water under continuous stirring at 90 °C for 2 h. The supernatant was filtered using a vacuum pump and stored at 4 °C in the dark. For the I-TiO_2_ and M-TiO_2_ preparation, two routes were followed: one using isopropyl alcohol and the other using methanol. A 0.1 M solution of titanium (IV) isopropoxide (Ti [OCH(CH_3_)_2_]_4_), TTIP, 99.5% (Sigma Aldrich; St. Louis, MO, USA) was added to 75 mL of the solvent. Next, 25 mL of the extract was added dropwise to the precursor solution and stirred for 4 h. After 24 h, brownish-green gel-like materials were obtained. The materials were dried at 80 °C for 8 h. The samples were then washed several times with deionized water to eliminate impurities. Finally, the obtained powders were calcined at 500 °C for 4 h in an air atmosphere. The powders were labeled as I-TiO_2_ and M-TiO_2_, indicating the solvent used.

### 4.2. Characterization of Green-Synthesized I-TiO_2_ and M-TiO_2_

The crystal structures of the I-TiO_2_ and M-TiO_2_ were characterized by X-ray diffraction (XRD) using an Empyrean diffractometer (PANalytical) equipped with a copper anode (l = 1.5406 Å). XRD patterns were obtained for 2θ ranging from 10 to 70° with a step size of 0.01°. Attenuated total reflection Fourier transform infrared (ATR-FTIR) spectroscopy was used to investigate the presence of organic matter on the nanoparticle surface. ATR-FTIR spectra were recorded in the 4000–400 cm^−1^ range using an IR Affinity-1S spectrometer (Shimadzu). The optical properties were assessed using the absorption spectra obtained using a Cary 5000 UV–Visible (UV–Vis) spectrometer (Agilent Technologies) equipped with a polytetrafluoroethylene (PTFE) integration sphere. The absorbance spectra were recorded in the range of 200–800 nm. The morphology of the NPs was investigated using field-emission scanning electron microscopy (FESEM) (TESCAN MIRA3 model). The Brunauer–Emmett–Teller (BET) method was used to determine the specific surface area (SBET) using a Bel-Japan MiniSorp II instrument. The particle size and zeta potential of the liquid suspension (1 mg/mL) were measured at 25 °C using a Zetasizer Pro instrument (Malvern Instruments).

### 4.3. HPLC Analysis

The chemical composition of the isopropanol and methanol extracts from blueberry was recorded by HPLC. Briefly, extracts were analyzed on an Agilent Technologies 1200 series instrument integrated with a DAD detector. The selected column was an RP-18 Zorbax (150 mm × 4.6 mm, 3.5 mm). The mobile phases consisted of water acidified with formic acid (A) and acetonitrile (B). The analysis was executed by injecting 10 mL of the extract and following the polarity gradient as previously published by our research group [39], with minor modifications regarding the selected time. Still, the absorbance of compounds was investigated at 254 and 365 nm. The identification of compounds was performed considering commercially available standards such as apigenin, catechin, hesperidin, isoharmentin, kaempferol, morin, myricetin, naringenin, phloretin, phlorizin, quercetin, and rutin. The following phenolic acid standards were also utilized: 3,5-dihydroxybenzoic, β-resorcillic, chlorogenic, ferulic, gallic, *p*-coumaric, *p*-hydroxybenzoic, proto catechuic, rosmarinic, synapic, and syringic/gentisic and vanillinic acids.

### 4.4. Analysis of Compatibility of Green-Synthesized I-TiO_2_ and M-TiO_2_ with PGPMs

#### 4.4.1. Isolation and Culture of PGPB

*B. subtilis* (ATCC 6633), *B. thuringiensis* (B-BT0001), and *T. harzianum* (F-BT0002) were isolated from soil and characterized and stored at the Tecnológico de Monterrey Ceparium. These strains were grown in Luria-Bertani (LB) broth or Potato Dextrose Agar (PDA) and incubated at 30 °C with continuous shaking at 180 rpm overnight as a precursor culture for the biocompatibility liquid and solid assays.

#### 4.4.2. Spread Plate Method

CFU counting was performed following the double-layer assay [37]. The Petri dish experiment consisted of two layers: the first layer (Bottom) was made of LB agar medium only, and the second layer (Top) was supplemented with I-TiO_2_ and M-TiO_2_ NPs at concentrations of 50, 100, and 150 μg/mL. Subsequently, 100 μL of a 10^−6^ dilution of the overnight-grown cultures of *B. thuringiensis* (B) or *B. subtilis* (Bs) was spread using a sterile bent glass rod. Using an appropriate formula (Equation (1)), CFU/mL and colonies were counted after an incubation period at a temperature of 30 °C. Kanamycin-containing Petri dishes were employed as a negative control, and two layers of LB agar Petri dishes were used as a control. The experiment was conducted in triplicate to ensure statistical and experimental reliability.(1)$$\frac{CFU}{mL}=\frac{\left(Number\;of\;colonies\times Dilution\;factor\right)}{Volume\;of\;culture\;plated}$$

#### 4.4.3. Fungal Compatibility

The compatibility between TiO_2_-NPs and *T. harzianum* was assessed by inoculating a 0.5 cm square of fungal mycelium onto PDA supplemented with 50, 100, and 150 μg/L of I-TiO_2_ or M-TiO_2_ NPs. The plates were incubated at 30 °C for six days until mycelial growth and sporulation were observed. Then, 5 mL of sterile distilled water was added to the agar surface, and spores were gently detached with a brush, taking care not to disturb the mycelium. The resulting suspension was collected in a 50 mL Falcon tube. Fungal spore counts were performed using a hemocytometer (Neubauer, Grid Optik) following standard procedures [37]. The data were expressed as spores/mL according to Equation (2). Terbinafine served as a positive control for the tests.(2)$$\frac{Spores}{mL}=\frac{Number\;of\;spores\times 10^{4}\times Dilution\;factor}{Number\;of\;squares}$$

### 4.5. Effect of Green-Synthesized I-TiO_2_ and M-TiO_2_ and PGPMs in C. annuum

Seeds of the Serrano pepper were sourced from local stores in Zapopan, Jalisco, Mexico. The seeds were disinfected with 70% ethanol and then washed twice with distilled water to remove potential impurities. The seeds were sown in seedling beds containing a sterilized substrate made of Sunshine^®^ Mix 3 and Perlite in a 3:1 ratio. Once the germination phase was complete and the seedlings reached a height of 10–15 cm, they were transferred to a greenhouse environment. The roots were then inoculated with different treatments consisting of I-TiO_2_ and M-TiO_2_ NPs with or without PGPMs (B, T, Mcg). The experiment involved 20 seedlings per treatment, with the greenhouse environment maintained at an average humidity of 35–35% and a temperature range of 25–35 °C. After 15 days, a second application of the treatment was administered, with a total incubation duration of 60 days. Upon completion, the growth of shoot and root lengths from their intersection to their tips was meticulously measured using a vernier caliper. The other physiological parameters include the number of leaves as well as fresh and dry weight. After the measurements, twelve seedlings were frozen using liquid nitrogen and preserved at −80 °C for further biochemical analysis. Additionally, eight fresh seedlings were randomly selected and weighed on an analytical balance to determine the fresh weight for each treatment. The seedlings were then wrapped in aluminum foil and subjected to a drying process at 100 °C for 2 h, after which the dried weight was recorded [40].

### 4.6. Analysis of Biochemical Features

#### 4.6.1. Total Phenolic Compounds Assay

A quantitative total phenolic amount was calculated following the Folin–Ciocalteu reagent (FCR) method as published [37]. The sample (seedlings) was crushed into a fine powder using liquid nitrogen and subsequently employed for the extraction of total phenolic contents. A total of 35 mg of the sample was gently homogenized using a solvent mixture (formic acid, methanol, and water—3:24:25 in volume). Thereafter, the homogenized samples were sonicated for 60 s using an amplitude of 80%, shaken for 20 min at 200 rpm, and centrifuged for 15 min at 4000 rpm. The experiment was followed on a spectrophotometer at 750 nm, based on the gallic acid standard curve (0–70 µg/mL), and total phenolic contents were calculated per gram of fresh weight.

#### 4.6.2. Total Protein and POX Activity

A crude extract was prepared in a 50 mM sodium phosphate buffer, centrifuged, and used for total protein quantification and measurement of enzyme activity. The Bradford method was used to quantify total protein, with serum albumin for the standard curve (0–50 µg/mL) [29]. For the evaluation of peroxidase (POX) activity, an experiment was conducted in a 96-well microplate [37]. A total of 10 µL of plant extract (35 mg powder/mL) was combined with 20 mM guaiacol as the substrate, and the reaction was initiated with the addition of 60 mM H_2_O_2_ at room temperature, yielding a final reaction volume of 250 µL. The enzymatic activity was monitored spectrophotometrically, and the change in absorbance by one unit per minute because of guaiacol oxidation was defined as a single unit of peroxidase activity (1 UA). This activity was normalized and expressed as units per milligram of fresh weight (UA/mg FW).

#### 4.6.3. Total Chlorophyll Quantification

Evaluation of total chlorophyll (chlorophyll a and b) was adopted from the method specified in [41]. For each sample, 0.1 g of plant leaf tissues was homogenized in 10 mL of 80% acetone (*v*/*v*) to extract chlorophyll pigments properly, followed by centrifugation at 4000 rpm for 20 min, and the resulting supernatant was carefully collected for analysis. Thereafter, the supernatant was transferred to a 96-well microplate (Agilent Biotek Synergy/HTX Multimode Reader, Santa Clara, CA, USA), and absorbance readings were taken at 663 nm and 646 nm using a microplate reader. As a result, chlorophyll a (C_a_), chlorophyll b (C_b_), and total chlorophyll (C_t_) content were calculated using Equations (3) and (4). Total chlorophyll was reported as the sum of chlorophyll a and b, expressed in μg per mL of plant extract.(3)$$Ca=12.21AA_{663}-2.81A_{646}$$(4)$$Cb=20.13A_{646\;}-5.03A_{663}$$

### 4.7. Statistical Analysis

Analysis of variance (ANOVA) was performed to assess the effects of nano-fertilization and bio-nano fertilization treatments. The necessary statistical assumptions for ANOVA were checked, including data normality (Shapiro–Wilk test), homogeneity of variances (Levene test), and independence of observations. All analyses were conducted using R software (R Core Team, 2025; Integrated Development Environment for R. Posit Software, PBC, Boston, MA, USA) within the RStudio environment (RStudio Team, 2025). The dataset comprises bacterial and NP compatibility, physiological, and biochemical responses. A Tukey’s post hoc test was used to identify specific pairwise differences between treatments. Partial least squares regression (PLSR) was applied to evaluate the relationship between the physicochemical properties of TiO_2_-NPs (BG, PS, BET, ZP, and DLS) and the physiological and biochemical responses of plants (height, number of leaves, fresh and dry biomass, total phenols, chlorophyll, peroxidase activity, and total protein). The predictor and response matrices were autoscaled before fitting. The model was constructed using a PLS algorithm with leave-one-out (LOO) cross-validation to select the optimal number of components. VIP values, as well as scores and loadings, were calculated to determine the descriptors with the greatest contribution and the grouping of treatments in the latent space. The analyses were performed in RStudio using the pls package.

## 5. Conclusions

This work reported a comparative analysis regarding the capacity of green-synthesized I-TiO_2_ and M-TiO_2_ in the biocompatibility of PGPMs and the promotion of physiological and biochemical responses among *C. annuum* cultivars. I-TiO_2_ and M-TiO_2_ were synthesized using isopropanol and methanol-based extracts from blueberries. The HPLC-DAD analysis demonstrated that extracts are mainly constituted by rutin, quercetin, hesperidin, and dihydroxybenzoic acid, which can be associated with the formation of the obtained I-TiO_2_. Still, it is important to note that according to the implemented method, the utilized commercial standards used in this work were partially identified in the isopropanol extract, for which further approaches such as HPLC-MS/MS or LC-MS studies are required for determining, in detail, the phytochemical composition of the methanol extract. The characterization of I-TiO_2_ and M-TiO_2_ demonstrated that, despite the differences between the polarity of the utilized extract, it exhibited nanocuboid-like morphologies, characteristic absorption peaks located between 320–420 nm, and distinctive bands within the FT-IR region (3500–500 cm^−1^) related to the stretching absorption of O-H, Ti-O, or Ti-O-Ti groups. The determined size of I-TiO_2_ and M-TiO_2_ was 146.10 and 546.80 nm, respectively.

Both formulations presented the expected characteristics of the anatase phase and stable colloidal behavior, although they showed differentiated biological responses depending on the type of synthesis. At low and moderate doses (50–150 μg mL^−1^), TiO_2_-NPs increased leaf number and plant height; in the case of I-TiO_2_, a consistent increase in fresh and dry biomass was observed. Biochemically, peroxidase activity and total phenolic content were elevated, especially when the NPs were combined with individual PGPMs, supporting a model in which anatase-induced redox signals activate antioxidant and phenolic pathways. Furthermore, total chlorophyll content increased, with particularly high doses of I-TiO_2_, consistent with increased photosynthetic efficiency and light harvesting. The identity of the PGPM modulated the nano-bio response: combining TiO_2_-NPs with *B. subtilis* or *T. harzianum* amplified the positive effects, whereas microbial consortia (Mcg) showed less consistent results, likely due to metabolic competition between strains or interference in colonization. Surface chemistry and electrostatic stabilization help explain the differences between formulations: the more negative ζ-potential of M-TiO_2_ suggests greater colloidal stability and overall compatibility, whereas I-TiO_2_ more markedly influenced biomass and photosynthetic parameters. Overall, the results indicate that (1) dose and formulation determine a biphasic response, (2) co-application with a single PGPM is more effective than the use of consortia under the conditions tested, possibly associated with microbial competition, and (3) green synthesis provides surface functionalities that promote dispersion, microbial compatibility, and predictable physiological responses. The nanoparticle formulation with the highest colloidal stability (I-TiO_2_) provides the best agronomic outcomes, revealing colloidal dispersion as a central design principle for next-generation nano-bio stimulants. Finally, as noted in the performance of I-TiO_2_, the retrieved evidence from this work evidenced the influence of extract polarity in the capacity of metal-based NMs, with beneficial effects on the improvement of physiological and biochemical responses.

## Figures and Tables

**Figure 1 plants-14-03672-f001:**
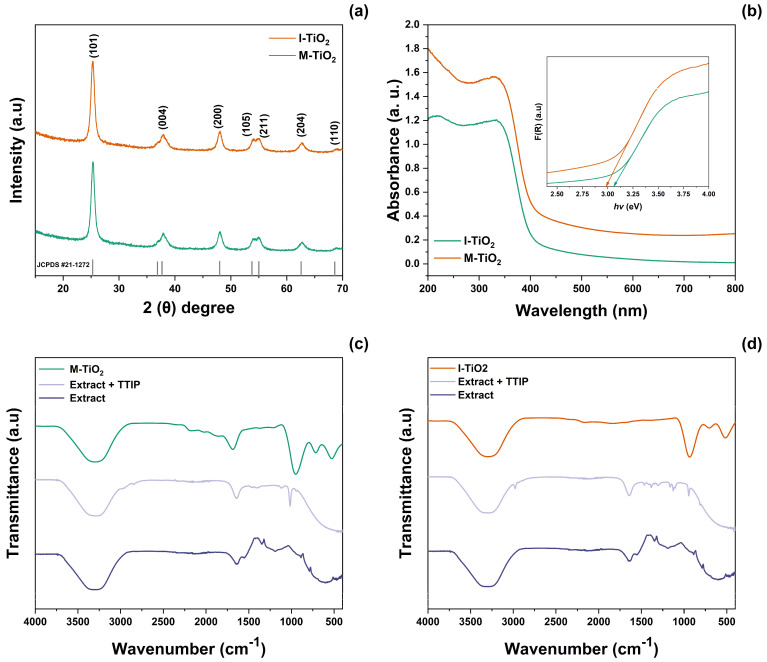
(**a**) XRD patterns and (**b**) UV–Vis spectra of the M- and I-TiO_2_-NPs. The FT-IR spectra of the (**c**) M- and (**d**) I-TiO_2_-NPs show the spectra of the blueberry extract, precursor solution, and calcined sample.

**Figure 2 plants-14-03672-f002:**
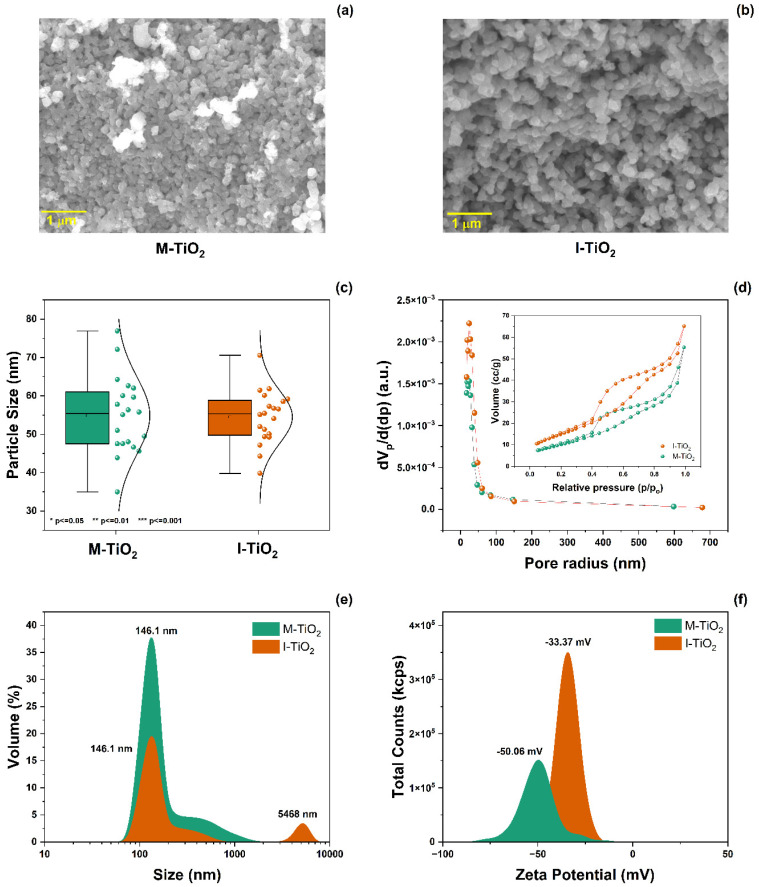
(**a**,**b**) SEM images, (**c**) grain size distribution, (**d**) pore size distribution (inset: N2 adsorption/desorption isotherms), (**e**) particle size distribution from dynamic light scattering analysis (DLS), and (**f**) zeta potentials of the M- and I-TiO_2_ nanomaterials.

**Figure 3 plants-14-03672-f003:**
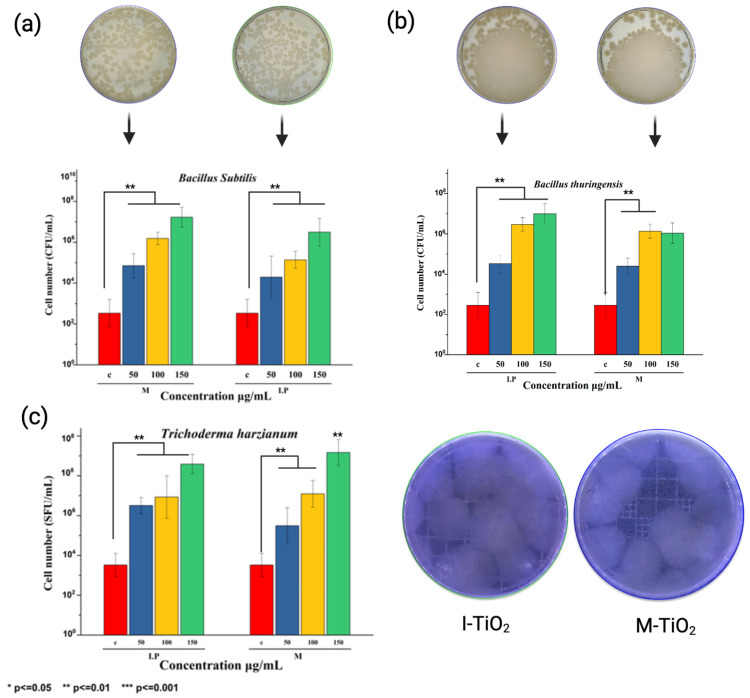
Microbial compatibility against M- and I-TiO_2_-NPs. The growth of (**a**) *B. subtilis*, (**b**) *B. thuringiensis*, and (**c**) *T. harzianum* was evaluated at concentrations of 0, 50, 100, and 150 μg/mL. The results are expressed as cell number (CFU/mL or SFU/mL). Statistical differences obtained by ANOVA and Tukey’s test are indicated by horizontal bars (* *p* < 0.05; ** *p* < 0.01; *** *p* < 0.001).

**Figure 4 plants-14-03672-f004:**
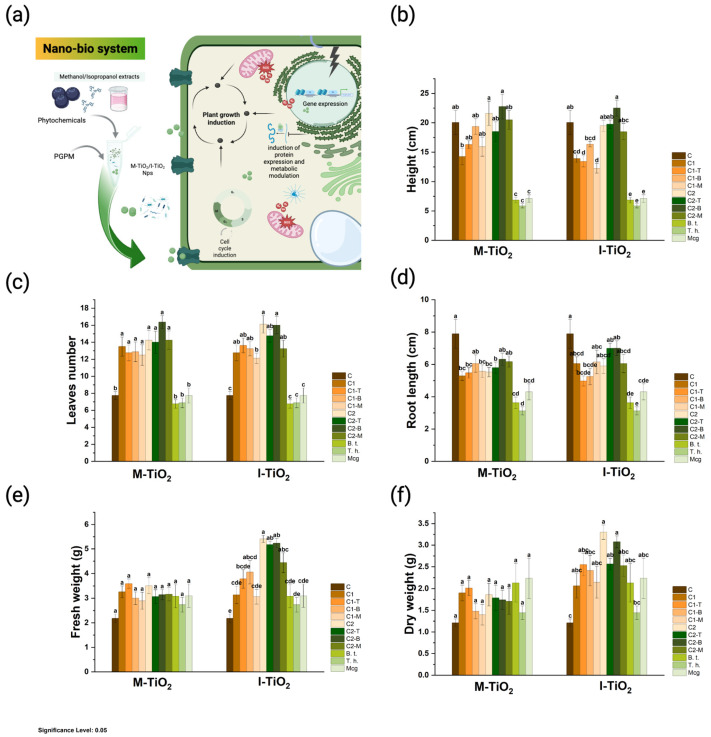
Growth responses of plants treated with M-TiO_2_ or I-TiO_2_. (**a**) representation of the activity of the reported nano-bio system; (**b**) height (cm); (**c**) leaves number; (**d**) root length (cm); (**e**) fresh weight (g); (**f**) dry weight (g); Treatments: C = control; C1, C2 = NPs alone; –T = with *T. harzianum*; –B = with *B. subtilis;* –M = with consortium (*B. subtilis* + *T. harzianum*). Different letters above the bars indicate significant differences (Tukey, α = 0.05) within each nanoparticle type (M-TiO_2_ or I-TiO_2_).

**Figure 5 plants-14-03672-f005:**
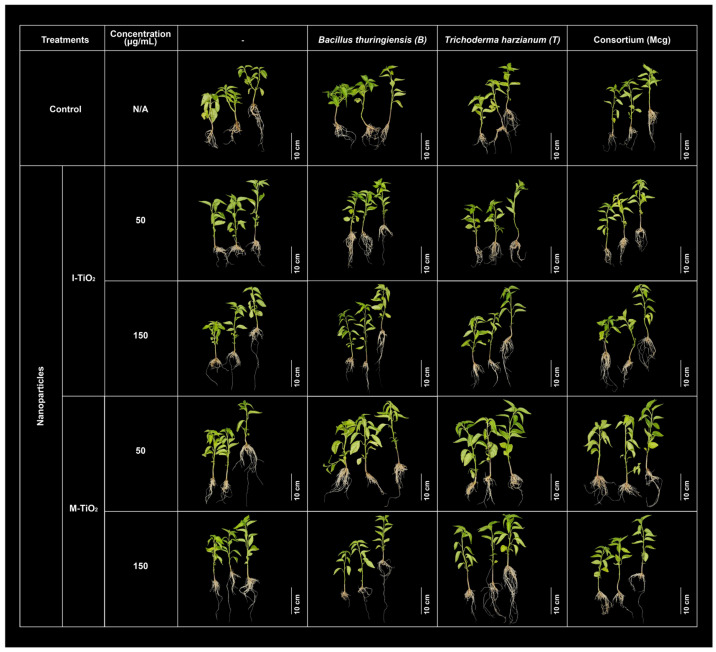
Plant growth upon exposure to TiO_2_-NPs synthesized with methanol (M-TiO_2_) or isopropanol (I-TiO_2_) at 50 and 150 μg mL^−1^, applied alone or with *B. thuringiensis*, *T. harzianum*, or a consortium (Mcg), compared to the control; N/A, no applicable.

**Figure 6 plants-14-03672-f006:**
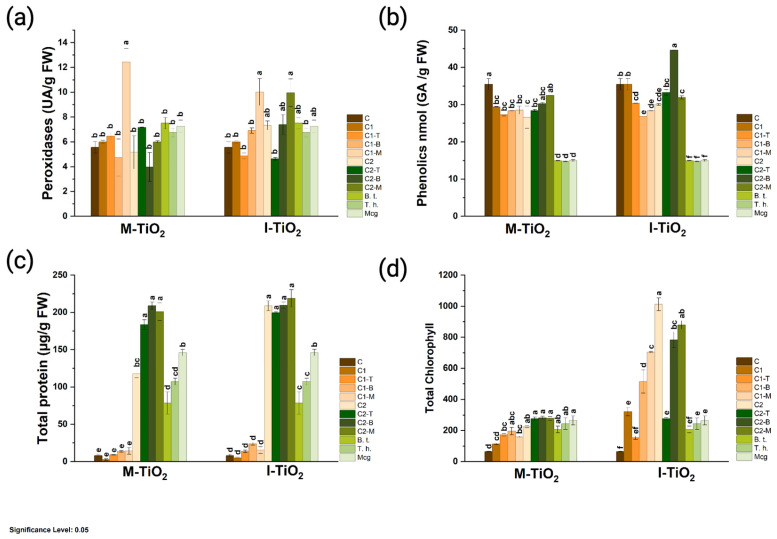
Biochemical response of plants treated with M-TiO_2_ or I-TiO_2_-NPs at two concentrations (50 and 150 μg/mL), applied alone or in combination with *B. subtilis* (B), *T. harzianum* (T), or a microbial consortium (Mcg). (**a**) Peroxidase activity, (**b**) total phenolic compound content, (**c**) total protein, and (**d**) total chlorophyll. Different letters indicate significant differences between treatments (Tukey, *p* < 0.05).

**Figure 7 plants-14-03672-f007:**
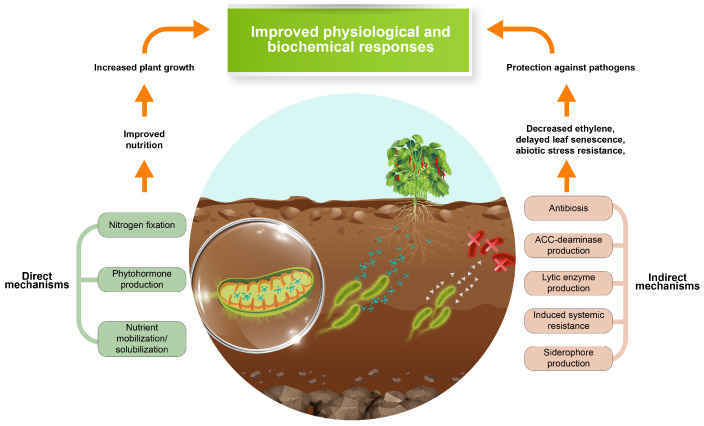
Representative scheme of potential multifactor effect of the nano-bio system in plants. Direct effect through improved nutrient absorption and promotion of biostimulant effectors by PGPM. Indirect effect as an immunomodulator of stress response.

**Table 1 plants-14-03672-t001:** HPLC-DAD analysis of the isopropanol extract from blueberry.

Retention Time [min]	Area [mAU*s]	Amt/Area	Amount [mg/mL]	Compound Name
5.854	14.89922	3.71612 × 10^−4^	5.53673 × 10^−3^	Rutin
12.936	13.29509	3.90834 × 10^−4^	5.19618 × 10^−3^	Quercetin
8.421	43.36256	2.83967 × 10^−4^	1.23135 × 10^−2^	Hesperidin
10.807	11.71115	1.94160 × 10^−4^	2.27384 × 10^−3^	3,5-dihydroxybenzoic

## Data Availability

The data presented in this study are available on request from the corresponding author. The data are not publicly available due to since all the information generated from this work is presented here.

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
