# Peer review of "From Bench to Greenhouse: The Comparative Nano-Bio System Effects of Green-Synthesized TiO2-NPs and Plant-Growth-Promoting Microorganisms in Capsicum annuum"

_plants, 2025, doi:10.3390/plants14233672_

Round 1
Reviewer 1 Report
Comments and Suggestions for Authors
Strongly recommend rejecting the submitted article, as the same authors have recently published the same research in the journal Antioxidant (https://doi.org/10.3390/antiox14060707) with only a few changes, resulting in a lack of novelty and interest in the application of nanotechnology in agriculture.
The current situation and growing interest in nanotechnology applied to the agricultural industry increase the requirements of scientific journals such as Plants to accept research papers with a high degree of novelty based on the development of advanced nanomaterials and not just trials with previously described and well-known nanomaterials such as the present article.
Author Response
Strongly recommend rejecting the submitted article, as the same authors have recently published the same research in the journal Antioxidant (https://doi.org/10.3390/antiox14060707) with only a few changes, resulting in a lack of novelty and interest in the application of nanotechnology in agriculture.
The current situation and growing interest in nanotechnology applied to the agricultural industry increase the requirements of scientific journals such as Plants to accept research papers with a high degree of novelty based on the development of advanced nanomaterials and not just trials with previously described and well-known nanomaterials such as the present article.
Dear reviewer, we sincerely appreciate your time and effort in reviewing our manuscript and for sharing your perspective. We understand your concern regarding the potential overlap with our previously published work in Antioxidants (https://doi.org/10.3390/antiox14060707). However, we would like to respectfully clarify that the present study differs substantially in both objectives and methodology.
In the Antioxidants paper, the nanoparticles were chemically synthesized using the molten salt method, while in the current work we employed a green synthesis approach using blueberry extracts as reducing and stabilizing agents. As you know, the synthesis route profoundly affects the morphology, crystallinity, surface chemistry, and band gap of nanomaterials, which in turn influences their biological interactions.
In this manuscript, we systematically compared two different blueberry solvent extracts, which generated distinct nanostructures and biological responses in Capsicum annuum. These findings highlight the importance of green synthesis in modulating plant physiological and molecular responses, representing a different scope and contribution to the field.
We acknowledge that advancing nanotechnology applications in agriculture requires a high degree of innovation, and we are confident that this work provides new insights into eco-friendly synthesis routes and their biological implications under greenhouse conditions, which—as you are aware—represents a critical step toward applied validation.
Once again, we appreciate your comments. Although the specific concerns raised do not point to aspects we can review directly, we have carefully considered feedback from all reviewers and incorporated the suggested changes (highlighted in red) to strengthen the quality and clarity of our manuscript.
Reviewer 2 Report
Comments and Suggestions for Authors
In the study, titanium dioxide nanoparticles (TiO2-NPs) were synthesized using blueberry extracts and, together with PGPM, were applied and studied for their combined effect on Capsicum annum functions and productivity.
In recent times, studies have widely investigated the effect of bio-based nanoparticles for promoting physiochemical traits in plants, overall yield and productivity, an important aspect of sustainable practices in agriculture and environmental well-being.
While such studies have shown both success and limitations, advanced biotechnological measures are being adopted to address the challenges with nanoparticle applications in agriculture.
Some suggestions and queries for improvement are provided.
Abstract: line 42-44: III) single PGPM co-application outperforms consortia…conditions. In the study, microbial consortia (Bacillus subtilis (Bs), Bacillus thuringiensis (B), and Trichoderma harzianum (T)) were co-applied with NPs; one of the species showed a better effect than the combined application. Further studies are required to assess the PGP effects of microbial species for additional development.
In addition, it is crucial to determine the compatibility of microbial strains for co-application so that the maximum favorable impact on the plant species is achieved.
Line 32-39: In greenhouse experiments, …………………………..with or without PGPM. Different results were obtained: I-TiO2 with a single microbial species increased fresh and dry plant biomass, while at higher doses, antioxidant response, an increase in total chlorophyll, and total protein content were achieved.
Further studies are required to optimize a nano-based system to achieve maximum plant growth and yield. According to the authors, what is the best nano system to achieve the desired results? What inference can be drawn from the study?
Line 105-108: This comprehensive approach seeks to elucidate……. environmental responsibility.
While the synthesis and application of bio-based nanoparticle systems have shown promising outcomes in multiple studies (focused on plant growth and productivity), further research is still required to establish feasible nano systems. What are the challenges in the large-scale application of nano-based systems? How can it be addressed? Discuss.
Material and Methods:
It is important to mention the details of the plant samples used, reference to the protocols followed, details of the instruments used, manufacturer, etc. No reference to the studies has been discussed.
Line 579-582: The tentative identification of compounds………….similar chromatography-based approach. What were the reference/standards used for compound identification? A reference citation to the published studies is required.
Line 636-638: 4.6. Analysis of Biochemical features
4.6.1. total phenolic compounds assay. A quantitative total phenolic amount………FCR method. A reference to the FCR method is required in the section, and likewise for other protocols followed.
Figure 1. A) XRD patterns and b) UV-Vis spectra of the M- and I-TiO2 materials. A reference and discussion to Figures C and D are required.
Title: In the title, it is advised to use the complete form for clarity; it is better to avoid abbreviations, e.g., PGPM
Please be consistent with the use of scientific names. In addition, it is suggested to use the complete forms at the first mention, then abbreviations may be used.
Round 2
Reviewer 1 Report
Comments and Suggestions for Authors
The current work lacks novelty in the development of new nanomaterials and their application. In addition, the same authors have recently published a similar study with the same nanoparticles that shows the synergistic effect in the strains mentioned.
Author Response
In the revised version, we incorporated an explicit structure–function statistical analysis to link the physicochemical properties of the green-synthesized TiOâ‚‚ nanoparticles with their biological performance. Specifically, we built a treatment-level dataset including bandgap energy (Eg), specific surface area, pore diameter, hydrodynamic size and zeta potential, together with plant physiological (height, leaf number, fresh and dry biomass) and biochemical responses (peroxidase activity, total phenolics, protein, chlorophyll. We then performed multivariate (PLS) analyses to identify which material properties explain most of the variance in plant and microbial responses. This multilevel structure–function analysis was not included in our previous Antioxidants paper and, in our opinion, clearly demonstrates that the present work goes beyond a simple repetition of biological assays (as you suggest) and much more importantly, we reveal the structural differences of the materials, which, upon closer inspection, are very different. Therefore, it is evident that the present work adds to the knowledge of understanding the function of nanoparticles and, beyond that, their structural property when forming a nano-biosystem providing a quantitative relationship for precision agriculture. We also include a table showing the different structural properties of the various nanoparticles.
Reviewer 2 Report
Comments and Suggestions for Authors
In the present form, the manuscript may be considered for publication.
Author Response
Dear reviewer, thank you for your comments and time that enables us to improve the quality of our work.